# Regulating the p53 Tumor Suppressor Network at PML Biomolecular Condensates

**DOI:** 10.3390/cancers14194549

**Published:** 2022-09-20

**Authors:** Magdalena C. Liebl, Thomas G. Hofmann

**Affiliations:** Institute of Toxicology, University Medical Center Mainz, Johannes Gutenberg University, 55131 Mainz, Germany

**Keywords:** biomolecular condensates, liquid–liquid phase separation, DNA damage, p53, PML, SUMO, nuclear body, cell death, cellular senescence

## Abstract

**Simple Summary:**

The cell nucleus is organized into different sub-nuclear compartments to control specific cellular processes, including PML nuclear bodies (NBs), also termed PML biomolecular condensates. PML-NBs form multi-protein complexes that are highly responsive to cellular stress, and regulate cell fate decisions in response to genome damage. By concentrating proteins that directly control the activity of the tumor suppressor p53, PML biocondensates guide the cellular response either towards induction of cell death or senescence through altering p53 modifications. In this review, we discuss the molecular mechanisms controlling PML condensate formation, and how they impact the regulation of p53 activity, and propose a list of promising candidate proteins that may contribute to the regulation of p53 at PML biomolecular condensates.

**Abstract:**

By forming specific functional entities, nuclear biomolecular condensates play an important function in guiding biological processes. PML biomolecular condensates, also known as PML nuclear bodies (NBs), are macro-molecular sub-nuclear organelles involved in central biological processes, including anti-viral response and cell fate control upon genotoxic stress. PML condensate formation is stimulated upon cellular stress, and relies on protein–protein interactions establishing a PML protein meshwork capable of recruiting the tumor suppressor p53, along with numerous modifiers of p53, thus balancing p53 posttranslational modifications and activity. This stress-regulated process appears to be controlled by liquid–liquid phase separation (LLPS), which may facilitate regulated protein-unmixing of p53 and its regulators into PML nuclear condensates. In this review, we summarize and discuss the molecular mechanisms underlying PML nuclear condensate formation, and how these impact the biological function of p53 in driving the cell death and senescence responses. In addition, by using an in silico approach, we identify 299 proteins which share PML and p53 as binding partners, thus representing novel candidate proteins controlling p53 function and cell fate decision-making at the level of PML nuclear biocondensates.

## 1. Introduction

The crowded and heterogeneous intracellular environment, which resembles a molecular jungle, requires cells to spatially organize their components to ensure the specificity and efficacy of functional interactions and enzymatic reactions in order to optimize cellular processes. The generation of macromolecular functional bio-entities is termed cellular compartmentalization, and is achieved by the establishment of numerous organelles that can be either membrane-delineated, including the endoplasmic reticulum, peroxisomes, and mitochondria, or membrane-less. Examples of membrane-less organelles in eukaryotic cells include nucleoli, Cajal bodies, and PML nuclear bodies (PML NBs) in the nucleus, as well as stress granules (SGs) and P-bodies in the cytoplasm. These membrane-less organelles are also termed biomolecular condensates, due to their function of selectively concentrating proteins and nucleic acids in a defined space [1,2]. In addition to concentrating biomolecules enabling efficient biochemical and cellular reactions, such condensates can also suppress biochemical reactions by sequestering or actively recruiting specific factors [3].

In recent years, it has become clear that many membrane-less organelles are formed along the principle of liquid–liquid phase separation (LLPS). Phase separation is a prominent term in modern material sciences and describes the process in which a homogenous liquid phase separates (or unmixes) into two or more compositionally distinct liquid phases [4]. A classic example of LLPS is the “demixing” of oil and water. LLPS leads to the formation of liquid droplets, whose liquid-like phase boundary allows the selective passage of specific molecules, but not others, thus facilitating the enrichment of those factors in the droplets compared to the bulk solution [5]. The compartments formed by LLPS exhibit a dynamic and rapid exchange of their components with the surrounding nucleo- or cytoplasm [6]. In recent years, different membrane-less cellular organelles have been shown to form according to the principles of LLPS, which is driven by specific physico-chemical features of proteins to allow their reversible unmixing. Interestingly, this process of biomolecular condensation is sensitive to cellular stress and is intimately linked to cellular stress responses and signaling chains, such as those triggered by cytotoxic and genotoxic stress [3,7].

The guardian of the genome, the tumor suppressor p53, is highly responsive to cellular stress and facilitates a wide range of cellular responses, such as DNA repair, cell cycle arrest, cellular senescence, and apoptosis. In addition to these canonical responses, p53 is involved in the regulation of metabolism, autophagy, ferroptosis (an iron-dependent form of cell death), stem cell maintenance, and the restriction of invasion and metastasis [8,9,10].

Although p53 receives strong scientific interest and is under intensive investigation, how p53 selects the appropriate cellular reaction in response to stress signals is not fully understood yet. In addition to temporal p53 expression dynamics and the interaction of p53 with numerous co-factors, post-translational modifications (PTMs) have key functions in shaping the p53 response and thus, p53-guided cell fate decisions [11]. p53 has been reported to be decorated by a plethora of different PTMs, with the phosphorylation of serine and/or threonine residues along with the acetylation, methylation, and ubiquitylation of lysine residues being the most frequently reported p53 PTMs. Other covalent modifications of p53 include SUMOylation, neddylation, UFMylation, OGlcNAcylation, ADP-ribosylation, hydroxylation, and β-hydroxylation [12,13]. The regulation of some of these p53 PTMs has been linked to PML NBs, where p53 has been shown to be recruited to, and to meet a subset of its modifiers influencing cellular life and death decisions upon stress [14].

In this review, we discuss in the context of LLPS the current knowledge-based view about the formation of PML NBs, the recruitment of p53 and its modifiers to PML NBs, as well as the role of PML NBs in p53 PTMs and the shaping of the p53 response. In addition, by following an in silico approach, we provide a list of novel candidate proteins regulating p53 function at PML biomolecular condensates.

## 2. The PML Nuclear Body and Its Functions

PML NBs are multi-protein complexes, for whose assembly, exclusively, the PML protein is essential [15,16]. PML is expressed in various isoforms, which share core protein modules and differ in some specific motifs [17]. PML derived its name due to the discovery that over 90% of patients with acute promyelocytic leukemia (APL) harbor a chromosomal translocation, which joins the PML gene with the gene encoding the retinoic acid receptor alpha (RARα) [18]. The resulting oncogenic fusion protein disrupts the structural integrity of PML NBs delocalizing them into countless micro-speckles and interferes with nuclear receptor signaling, thus inhibiting cell differentiation and driving leukemic carcinogenesis in these patients [19,20]. Therapy-induced degradation of PML-RARα by arsenic trioxide (As_2_O_3_), a drug originating from traditional Chinese medicine, and all-trans retinoic acid (ATRA) triggers cell death or terminal myeloid differentiation, respectively. Both responses are linked to the restoration of PML NB formation causing remission in over 95% of APL patients [21].

PML NBs are detected as discrete macromolecular foci in the nucleoplasm and rarely in the cytoplasm. In mammalian cells, 1–10 PML NBs per nucleus with a diameter of 0.2–1 µm are typically present. The number, size, and morphology of PML NBs are dynamic and change depending on the cell type, the cell cycle phase, as well as physiological and pathological stimuli, particularly cellular stress signals [22,23]. In contrast to other biomolecular condensates such as nucleoli and stress granules, which contain nucleic acids particularly RNA, the data for PML NBs on this topic are conflicting. Whereas some studies show that PML NBs contain RNA [24,25,26,27], others show that PML NBs are devoid of chromatin and RNA, but that nascent RNAs accumulate in the vicinity of PML NBs [28,29]. Though it is still under debate if PML NBs comprise nucleic acids, it is clear that they consist of multiple different proteins.

In addition to PML, over 160 proteins have been reported in association with PML NBs [30]. However, only a fraction of these factors has been detected at PML NBs under physiologically relevant conditions and endogenous protein levels. PML-associated proteins are involved in diverse cellular functions, including transcriptional regulation, cell cycle regulation, post-translational modifications, virus–host interactions, and DNA damage repair [22,30]. Thus, it is not surprising that biological processes regulated by PML NBs reflect the functions of the PML NB-associated proteins. PML NBs have been shown to be involved in numerous cellular processes, including protein modifications, especially protein SUMOylation [19]; gene expression and epigenetic regulation [31,32]; the DNA damage response [33,34,35,36]; apoptosis [37]; cellular senescence [38]; and antiviral responses [39]. Additionally, PML NBs have been linked to the regulation of nuclear protein availability by serving as storage depots for proteins. Upon specific stimuli, such as heat shock, cytokine signaling, or genotoxic stress, proteins can be released from PML NBs making them available when required [23,40,41,42,43]. Finally, PML NBs also play a role in protein degradation and protein quality control via the recruitment of SUMO-targeted E3 ubiquitin ligases [44,45], and by regulating the resolution of cytoplasmic stress granules [46,47], illustrating a molecular linker function between nuclear and cytoplasmic stress response pathways.

## 3. Biogenesis of PML Biomolecular Condensates

Since the PML protein is the only protein found to be required for PML NB formation [15], it is referred to as a scaffold protein, whereas all other PML NB-localized proteins are termed client proteins [1]. Interestingly, besides PML, the speckled 100 kDa protein Sp100, an anti-viral restriction factor, is the only known constitutive component present at PML NBs [22]. The PML protein forms a spherical shell with a thickness of 50–100 nm at the periphery of PML NBs, with other PML NB-associated proteins localized either in the core or the periphery of PML NBs [48].

### 3.1. The Phases of PML NB Formation

For simplicity, two phases can be distinguished during PML NB biogenesis. The initial phase involves self-polymerization due to the autointeraction between PML proteins into the ordered hollow spheres (Figure 1). This nucleation process is driven by covalent disulfide bridges between oxidized cysteine-residues of PML monomers as well as non-covalent interactions between the N-terminal RING finger/B-box/coiled-coiled (RBCC) region of PML [23,49,50,51], which is shared by all PML isoforms [17]. Specifically, it has been shown that the initial formation of PML NBs depends on RBCC oligomerization mediated by tetramerization of the RING domain and oligomerization of the B1 box of the RBCC region [51,52]. All PML isoforms have the intrinsic capacity to condense into PML NBs when transfected into PML knockout (KO) cells [53].

During the maturation phase, additional proteins are recruited to PML NBs through a uniform molecular mechanism using SUMO as a molecular glue (Figure 1). The proteins recruited are either covalently SUMO-decorated and/or contain one or more SUMO-interacting motifs (SIMs), which are about four-amino acids-short hydrophobic stretches that non-covalently bind SUMO proteins [54,55]. The RING domain of PML interacts with and recruits UBC9 (encoded by the UBE2I gene), the only known SUMO E2-conjugating enzyme in humans, which can either directly or with the help of SUMO E3 ligases catalyze the covalent attachment of SUMO to specific lysine residues, which are part of the consensus or non-consensus SUMO-modification motifs within its substrate proteins [56]. SUMO modifications can be reversed by SUMO-specific peptidases/proteases of the SENP family, such as SuPR-1/SENP2, which remove SUMO from PML [57]. PML itself is SUMOylated at three major SUMO conjugation sites, i.e., K65, K160, and K490 [58]. Additionally, PML has been found to be SUMOylated at K487 and K616 [59,60]. Employing mass spectrometry-based approaches facilitated the discovery of further SUMOylation sites of PML, namely K380, K400, K460, and K497, in recent years [61,62,63]. Although the detailed function of these modifications still remains elusive, they might regulate the recruitment of SIM-containing proteins to PML NBs.

Protein SUMOylation facilitates protein–protein interactions via non-covalent interactions between SUMOylated proteins and proteins containing SIMs, thereby creating a dense protein meshwork. PML harbors a C-terminal SIM, enabling interaction with SUMOylated proteins, including other PML proteins [64]. Although SUMO–SIM-mediated PML–PML interactions are dispensable for the initial PML NB formation [54], they might be important for PML NB architecture depending on the specific PML isoform. Upon transfection in PML KO cells, all PML isoforms except PML-II and -VI, which formed spherical structures, were able to assemble into toroidal, i.e., hollow, spherical structures. The introduction of mutations in the SIM affected the ability of the PML isoforms to form toroidal PML NBs differently: PML-I and PML-IV no longer generated toroidal structures, whereas there was no effect on the hollowness of the structures formed by PML-III and V. In contrast, mutation of the SIM allowed PML-VI to assemble into toroidal structures. [65]. Thus, different PML isoforms may assemble into differently shaped three-dimensional structures. Whether this also occurs under non-overexpression conditions and if this is also functionally meaningful for the biological responses regulated by PML NBs remains to be determined.

Undisputedly, intermolecular SUMO–SIM interactions play a crucial role in the recruitment of client proteins to PML NBs, and, thus, in the maturation phase. Most PML NB-associated proteins can be SUMOylated and/or harbor SIMs [66]. Interestingly, PML decorated with polySUMO chains composed of SUMO-2 and SUMO-3 serves as a degradation signal by facilitating the SUMO–SIM-mediated recruitment of the SUMO-interacting E3 ubiquitin ligases, RNF4 and RNF111, which mediate PML poly-ubiquitination and subsequent proteolytic (or non-proteolytic) ubiquitylation events [45]. Such PML turn-over may regulate or terminate PML-NB-regulated stress responses.

Using a proximity labeling mass spectrometry approach, 59 proteins were identified to interact with PML in a SUMO-dependent manner. Since this approach cannot discriminate between interactions mediated via covalent SUMOylation or non-covalent SUMO–SIM interactions [67], the detailed molecular mechanisms underlying individual interactions still remains to be determined. However, experiments with ectopically expressed proteins indicate that PML NBs generated by wild-type PML preferentially recruit poly-SIM-containing proteins, whereas PML NBs formed by PML lacking the three lysine residues, which serve as the main SUMOylation acceptor sites, attract poly-SUMO-proteins [68]. These results indicate that cells can control PML NB composition by regulating the SUMOylation/deSUMOylation of PML and client proteins. Body composition can also be fine-tuned by post-translational modifications of PML and/or the client proteins. Notably, the phosphorylation of residues adjacent to SIMs usually increases the affinity of the SIM-containing protein to SUMO, whereas the acetylation of SUMO typically weakens the SUMO–SIM interaction [69,70]. Recently, it has been shown that the interaction of a phosphomimetic PML SIM mutant with SUMO1 is blocked by the intrinsically disordered N-terminal region of SUMO1 (but not SUMO2). This inhibitory effect can be overcome by the addition of Zinc. This finding adds another layer of complexity on how cells can regulate PML NB biogenesis [71]. Thus, PML NBs are dynamically controlled biocondensates that change their composition and function through recruiting and expelling their client proteins.

### 3.2. LLPS Contributes to PML NB Biogenesis

In recent years, PML NB formation has been revisited in light of liquid–liquid phase separation (LLPS). PML NBs possess key properties of biomolecular condensates formed by LLPS, e.g., they generally have a spherical morphology, they can undergo fusion and fission events, and they exhibit dynamic component exchange with the surrounding nucleoplasm. LLPS occurs when a critical concentration threshold is crossed, which enhances weak protein–protein interactions [72]. PML NBs have been shown to be disassembled if the PML concentration is lower due to an expansion of the nuclear volume [73]. These results indicate that LLPS might play a role in PML NB biogenesis.

A key molecular driving force of LLPS are weak multivalent intra- or inter-molecular interactions between proteins and/or nucleic acids [3]. LLPS of proteins is determined by two types of protein architectures capable of forming multivalent interactions, i.e., proteins with intrinsically-disordered regions (IDRs) and proteins with modular domains [74]. IDRs are stretches of amino acids with low sequence complexity, meaning that certain amino acids are observed at a higher-than-expected ratio. Specifically, polar amino acids, such as glycine, serine, glutamine, proline, glutamic acid, lysine, and arginine, as well as the aromatic residues, tyrosine and phenylalanine, are enriched in IDRs. These IDRs pattern the polypeptide backbone, provide multiple interacting motifs or “stickers”, and enable weak interactions, thus promoting phase separation [7]. Many proteins residing in PML NBs, including PML itself, are predicted to contain IDRs [75,76].

Repetitive SUMO proteins (covalently attached to target proteins), which can be bound by SIMs, represent one type of modular interaction domain providing multivalent interactions and driving LLPS. The first experimental evidence for the involvement of LLPS in the biogenesis of PML NBs came from the observation that mixes of repetitive SUMO and SIM polymers condense in liquid-like droplets in vitro and in transfected cells. Depending on the ratio of non-bound SUMO:SIM sites in the polymer, either SIM- or SUMO-containing clients are recruited: if non-bound SUMO modules are in excess, SIM-containing clients partition into the droplets and vice versa [68]. These results suggest that LLPS is involved in PML NB maturation and client recruitment after the initial formation of PML polymers.

PML harbors several SUMOylation sites, as well as a SIM that may facilitate LLPS through intra- and/or inter-molecular multivalent SUMO–SIM interactions. Additionally, the PML protein contains a coiled-coil motif, which has been found in other proteins capable of LLPS, and its C-terminal disordered structure is typical of phase-separating proteins [5,76,77]. One characteristic property of biomolecular condensates is the rapid exchange of components with the surrounding environment, which can, for example, be investigated by fluorescence recovery after photobleaching (FRAP) [78]. Using this method, Fonin et al. found that PML NBs with a size below 1 µm^2^ and a generally spherical morphology show a rapid, dynamic exchange with the surrounding nucleoplasm, whereas bigger PML NBs display a predominantly toroidal structure with low exchange dynamics [76]. The dynamic behavior of specific PML isoforms warrants further investigation, since conflicting results of FRAP experiments using specific PML isoforms have been reported [53,79,80]. This might be partially caused by the use of different cell lines (human and murine), as well as cells with endogenous PML or PML KO cells. Phase transition properties of the PML protein have also been demonstrated by an in vitro LLPS assay. GFP-PML was found to form liquid droplets in the presence of a crowding agent, which promotes LLPS. This droplet formation was dependent on the concentration of GFP-PML, reflecting the requirement of phase-separating proteins to overcome the solubility limit, and the salt concentration, which affects multivalent interactions. Additionally, 1,6-hexanediol, a widely used disrupter of weak interactions and, thus, LLPS, led to the disappearance of the GFP-PML spherical droplets [81].

Although there is evidence for the involvement of LLPS in PML NB biogenesis, the experimental settings used to study LLPS are quite artificial and, in consequence, prone to generate artifacts. Thus, its exact function warrants further investigation and important questions need to be addressed in the future, such as: is LLPS only involved in the maturation phase of PML NB formation by recruiting client proteins via SUMO–SIM condensation, or are multivalent intra-molecular interactions between the predicted intrinsically disordered region of PML involved in the nucleation of PML NBs?

PML NBs exhibit an architecture of an inner core surrounded by a PML shell. Similar core-shell architectures have been also observed in other membrane-less organelles, such as nucleoli and stress granules [7]. It has been shown that the sub-compartments of nucleoli are formed through the immiscibility of different liquid phases [82]. It remains to be determined if a similar mechanism leads to the dual-phase architecture of PML NBs, and to what extent the shell and the inner core possess different biophysical properties. In contrast to nucleoli, where RNAs have been shown to be involved in biocondensate formation [83,84], RNAs apparently do not play a fundamental role in PML NB biogenesis, as transcription inhibition does not disrupt PML NBs [25,85]. Thus, PML NB formation seems to be driven by the oligomerization of PML and SUMO–SIM-dependent recruitment of associated proteins, with a likely contribution of LLPS to both processes.

## 4. PML Biocondensates and the p53 Response

Interactions between p53 and PML have been identified at different molecular levels. Interestingly, regulatory interactions between p53 and PML at the transcription level have been found. On the one hand, PML NBs have been shown to associate with the p53 encoding gene locus [86,87]; on the other hand, PML is a direct target gene of p53 [88] indicating a feedback loop between PML and p53 expression upon stress. Furthermore, on the protein level, p53 has been shown to directly interact with PML, and to co-localize with PML NBs [89].

p53 can be conjugated with SUMO-1, -2, or -3 at K386, which was shown to stimulate its transcriptional activity [90,91,92]. However, other studies failed to detect any effect of p53 SUMOylation on p53 transcriptional activation [93] or even found an inhibitory effect [94,95], which, at least partially, may reflect differences in the cell models and/or reporter constructs used in these studies. Thus, the functional relevance of this PTM on p53 activity remains unclear. Although it is tempting to speculate that p53 SUMOylation might play a role in the recruitment of p53 to PML NBs because of the importance of SUMO–SIM interactions for the association of client proteins to PML NBs, a p53 SUMO-defective mutant still localizes to SUMO1 foci, which likely represent PML NBs. Additionally, the isolated C-terminus of p53 (amino acids 294–393), which includes the SUMOylation side, was not recruited to PML NBs [96]. Together, these results suggest that p53 SUMOylation might be dispensable for p53 association with PML NBs. Thus, p53 localization to PML NBs seems to be driven by the direct interaction of its central DNA binding domain with the C-terminus of PML [97]. p53 recruitment to PML NBs is enhanced by MORC3, which forms independent nuclear domains, which associate with PML NBs via the SUMOylation of MORC3 and the interaction with the SIM of PML [98,99].

PML and PML NBs play an important role in p53 regulation. Under normal physiological conditions, p53 levels are kept low due to polyubiquitination and the subsequent proteasomal degradation of p53 [100]. MDM2 is the main E3 ubiquitin ligase of p53, and also localizes to PML NBs. At PML NBs, p53 was shown to interact with the death receptor TRAIL-R2 (TNFRSF10B), which leads to a reduction in p53 levels, potentially via the interaction of TRAIL-R2 with MDM2 [101]. In response to DNA damage, PML blocks p53 poly-ubiquitination and degradation by sequestering MDM2 to nucleoli [102]. This sequestration of MDM2 is counteracted by MAP kinase 7 (MAPK7) and the spindle-assemble checkpoint protein MAD1 (MAD1L1), which block the interaction between PML and MDM2 leading to p53 degradation [103,104]. PML NBs also contribute to p53 stabilization by recruiting the kinases, CHK2 (CHEK2) and CK1δ/ε (CSNK1D/E) in response to genotoxic stress, which phosphorylate p53 at S20 and S18, respectively [105,106,107]. These PTMs play a role in the disruption of the interaction of p53 with MDM2, thus leading to p53 accumulation [108].

Interestingly, p53 is recruited to PML NBs upon DNA damage induced by ionizing radiation, UV, or chemotherapeutic agents such as doxorubicin/Adriamycin and cisplatin [37]. At PML NBs, p53 meets many of its key regulatory enzymes including the protein kinase CHK2 [105,106]; the DNA damage-activated tumor suppressor kinase HIPK2 [109,110,111,112]; the acetyltransferases MOZ (KAT6A) [113], TIP60 (KAT5) [114], p300 (EP300), and CBP (CREBBP) [115]; and the deacetylase SIRT1 [116]. These modifiers fine-tune the p53 response by mediating post-translational modifications of p53, leading to either p53 activation or inactivation. In particular, the interplay between p53, HIPK2, SIRT1, and CBP/p300 at PML NBs has been studied in more detail and appears to be differentially modulated upon repairable versus irreparable DNA damage, which either results in the activation of p53’s apoptotic activity or its inactivation [117,118,119]. Thus, PML NBs constitute important regulatory protein biocondensates to modulate p53 phosphorylation and acetylation, thereby regulating p53 activity and cell fate decision-making. To what extent regulation of the p53 pathway is under control of LLPS-controlled protein recruitment mechanisms remains currently unclear. 

To discover new modifiers potentially interacting with p53 at PML NBs, we used an in silico approach and retrieved physical interactors of PML and p53 from the BioGRID database, restricting our analysis to human proteins [120]. This allowed us to identify 299 proteins (Figure 2a, Table 1) which interact with both PML and p53; among them, numerous previously mentioned enzymes (CHK2, HIPK2, MOZ, TIP60, CBP, and SIRT1), which are known to interact with p53 at PML NBs. Subsequently, gene ontology (GO) term enrichment was performed using the Database for Annotation, Visualization and Integrated Discovery (DAVID) [121]. GO analysis of the molecular function revealed that these 299 proteins are associated with DNA and RNA binding, as well as protein binding; specifically, ubiquitin and, as expected for p53-interacting proteins, transcription factor binding (Figure 2b). Furthermore, these proteins are linked to localization in PML NB bodies, as well as the nucleus and cytoplasm. This is in line with the observation that PML NBs are dynamic macromolecular structures that exchange their components, some of which are known to shuttle between the nucleus and cytoplasm, with the surrounding nucleoplasm.

The GO analysis also revealed that transcriptional regulation, protein SUMOylation, and the DNA damage response are among the top 10 significantly enriched biological processes. These biological processes are well-established to be regulated by PML and p53. Our approach identified a large set of potential p53 regulators that may modulate p53 function through association with PML NBs. We hope that this list of proteins may stimulate the reader to address potential unexplored molecular links of p53 with PML NBs in future analyses.

### 4.1. PML Biocondensates and p53 Post-Translational Modifications

p53 meets many of its key regulatory modifying enzymes at PML NBs, which regulate a set of important p53 post-translational modifications (Figure 3). Probably the best studied p53 modifier in association with PML NBs is the DNA damage-responsive kinase HIPK2. HIPK2 is activated upon genotoxic stress downstream of the DNA damage checkpoint kinases ATM and ATR, and interacts with p53 at PML NBs and phosphorylates p53 at S46 [109,110,111]. This modification of p53 is associated with cell death through transcription-dependent and -independent, mitochondria-associated effects. On the one hand, p53 S46 phosphorylation transactivates a distinct set of pro-apoptotic p53 target genes. On the other hand, this modification facilitates the binding of p53 to the pro-apoptotic Bcl2 family member BAX, stimulating BAX-dependent mitochondrial outer membrane depolarization and apoptosis induction [122]. A prerequisite for p53-dependent BAX activation is the isomerization of p53, which is catalyzed by the prolyl-peptidyl cis/trans isomerase PIN1 [123]. PIN1 mediates the dissociation of the apoptosis inhibitor IASPP (PPP1R13L) from p53 and stimulates p53 acetylation at K373 and K382 by the acetyltransferase p300 [124]. p53 K373/K382 acetylation, which enhances the transactivation of target genes, is also catalyzed by the acetyltransferase CBP. This process is stimulated by prior phosphorylation of p53 at S46 [109]. Interestingly, PIN1 reduces PML stability [125] and also mediates HIPK2 activation by contributing to HIPK2 stabilization upon DNA damage [126], reflecting a central role of this protein isomerase in the p53 pathway. Taken together, p53 S46 phosphorylation is important for mediating p53 acetylation at K382, two molecular modifications potentiated by HIPK2 [109].

In addition to p53 K373/K382 acetylation, acetylation of p53 at K120 by the acetyltransferases MOZ and TIP60 is also associated with PML NBs. Acetylation of p53 at K120 by TIP60 was shown to induce p53-dependent apoptosis [114,127,128], whereas MOZ-dependent p53 acetylation at K120 and K382 is associated with the induction of cellular senescence [113]. PML NBs are not only sites of p53 acetylation, but also regulate p53 deacetylation. The deacetylase SIRT1, which is recruited to PML NBs upon genotoxic stress, catalyzes the deacetylation of p53, promoting cell survival [119,129]. Upon irreparable DNA damage, SIRT1 activity at PML NBs is counteracted by HIPK2, which phosphorylates SIRT1 at S682, and inhibits SIRT1 deacetylase activity. Thereby, this inhibitory phosphorylation mark allows the efficient acetylation of p53 at K373/K382 and the induction of cell death upon cellular stress [119]. In summary, PML NBs concentrate p53 and many of its modifiers—both activators and negative regulators—and thereby constitute a macromolecular platform with a crucial role in the dynamic regulation of p53 phosphorylation, acetylation, and ubiquitylation. It will be interesting to see in the future to what extent post-translational modifications of p53 are regulated by LLPS-dependent mechanisms.

### 4.2. PML Biocondensates, p53 Downstream Responses, and Cancer

In addition to p53-mediated apoptosis, PML NBs also play a role in regulating p53-dependent senescence and oxidative stress response. Oncogene-induced senescence stimulates p53 phosphorylation and acetylation by CBP at PML NBs to induce cellular senescence [115,130]. Further evidence for the role of PML NBs in p53-mediated senescence comes from APL therapy. Therapy-induced restoration of physiological PML NBs promotes p53 activation and cellular senescence, which drive the therapeutic response [131]. Upon oxidative stress, PML NBs are crucial for p53 activation and the transactivation of anti-oxidant target genes [132]. Although the underlying molecular mechanisms are not clarified yet, concentrating p53 together with its modifying enzymes at PML NBs may contribute to specify the p53 response towards cell death, senescence, or cell survival. Taken together, PML NBs regulate p53-mediated cellular senescence, apoptosis, and the oxidative stress response.

PML and PML NBs are key mediators of p53 downstream tumor suppressive responses by regulating p53 stabilization, activation, and p53 PTMs, thus promoting p53-mediated apoptosis and cellular senescence upon DNA damage and aberrant oncogene signaling. Though the importance of PML NBs for wild-type p53 functions is undisputed, the role of PML NBs in regulating mutant p53 is currently unknown. p53 mutations, which are typically point mutations, are the most common mutations in human cancers, with 35–42% of tumors harboring p53 mutations [133,134]. These mutations lead to a loss of p53 wild-type functions, but also often confer oncogenic gain-of-function (GOF) properties [100]. Although it has been shown that a p53 nonsense mutant can still localize to PML NBs [135], raising the possibility that other p53 mutants, including the more common p53 missense mutations, also localize to PML NBs, the functional consequences remain largely unexplored. PML has been found to be important for cell proliferation, p53 transactivation [136], and STAT3 signaling in cancer cells harboring missense p53 mutations [137]. The interaction of mutant p53 with STAT3 has been shown to increase JAK2/STAT3 signaling, promoting cancer cell proliferation and tumorigenesis [138]. Thus, it might be possible that the association of mutant p53 with PML NBs mediates neomorphic GOF properties. Furthermore, it is also tempting to speculate that LLPS and/or deregulated LLPS events might contribute to the cellular phenotype and the carcinogenic function of cancer-derived p53 mutants.

### 4.3. PML Isoforms and p53

Although out of the seven investigated PML isoforms, all isoforms co-localized with p53 in PML NBs, only PML IV was able to stabilize p53, stimulate p53 S46 phosphorylation and K382 acetylation, and induce cellular senescence [139]. These observations are further supported by the findings that HIPK2 is recruited to PML NBs preferentially through interaction with PML IV, and also shows a phosphorylation-dependent interplay with PML. Through site-specific phosphorylation of PML at Ser8 and Ser38 upon DNA damage, HIPK2 stabilizes PML during the early phase of the DDR by stimulating PML SUMOylation [140]. Whether these mechanisms also contribute to the stimulative effect of HIPK2 on p53 activity remain to be determined.

Senescence induction by PML IV is dependent on the presence of a short amino acid stretch unique to PML IV. PML IV has been also shown to recruit ARF (CDKN2A) to PML NBs [141]. ARF plays a crucial role in p53 activation upon oncogene signaling by sequestering the p53 negative regulator MDM2 to nucleoli [108]. PML IV/ARF interaction enhances p53 SUMOylation by SUMO1, resulting in increased levels of the p53 target gene p21 (CDKN1A), a prominent inhibitor of cellular proliferation and activator of cellular senescence [141]. These results might, in part, explain why exclusively PML IV, and not the other PML isoforms, triggers p53-dependent cellular senescence [139].

## 5. Conclusions

During the last two decades, PML NBs have received strong scientific attention, and their important role in anti-viral response as well as in cell and cancer biology has been broadly documented. In particular, the function of these highly dynamic stress-responsive biomolecular condensates in regulating the p53 response and, therefore, cell fate decisions upon genome damage is still an attractive field of research warranting numerous interesting future findings.

Although the molecular determinants underlying the formation and disassembly and/or degradation of PML NBs is well understood, recent findings suggest an important contribution of LLPS to the establishment of PML biocondensates. Whether and how LLPS contributes to the biological effects exerted by the PML-p53 axis, including activation of the cell death and senescence response, and whether LLPS may also impact the recruitment of p53 modifiers to PML NBs remains to be elucidated. Finally, it will be interesting to find out whether such knowledge can be used to identify cancer-specific pharmacological regulators of LLPS, which may be used to stimulate the p53-driven cell death response in cancer.

To further stimulate the PML research field, in this review, we have distilled a list of 299 proteins previously shown to interact with both PML and p53, thus providing potential candidate proteins which may regulate p53 activity in association with PML nuclear condensates. We hope this will result in novel insights into the role of PML biocondensates in p53 regulation, and its link to LLPS-based mechanisms and cell fate control.

## Figures and Tables

**Figure 1 cancers-14-04549-f001:**
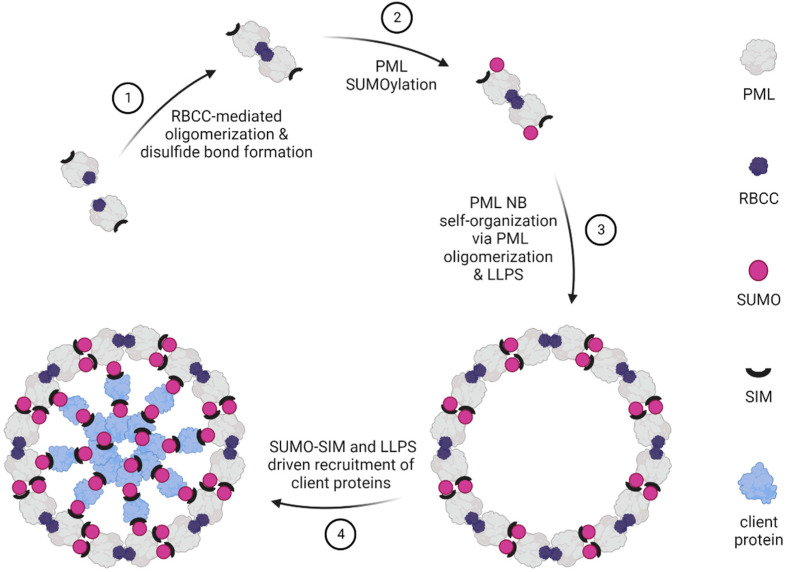
Model of PML nuclear body (NB) assembly. (1) PML NB biogenesis is triggered via oligomerization of PML monomers, which is mediated via oligomerization of the N-terminal RBCC region and disulfide bond formation between cysteines residues. (2) UBC9-dependent PML SUMOylation primes for multivalent intermolecular interactions between SUMO moieties and a SUMO interaction motif (SIM) present in the PML protein. (3) PML NB shell assembly is promoted through polymerization of PML via intermolecular non-covalent interactions and SUMO–SIM interactions. (4) SUMOylated and/or SIM-containing PML NB-associated proteins are recruited to PML NBs through SUMO moieties and SIMs present in PML and its client proteins. Liquid–liquid phase separation (LLPS) controls PML NB biogenesis via phase transition properties of the PML protein itself and SUMO–SIM interaction-driven phase transition.

**Figure 2 cancers-14-04549-f002:**
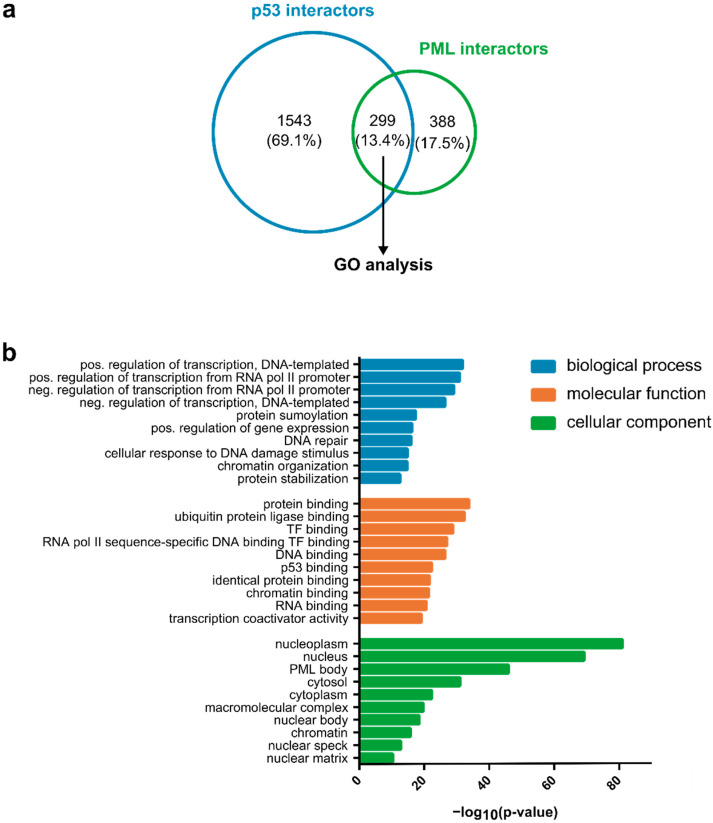
Identification of potential, functional protein interactions of p53 at PML biocondensates. (**a**) Overlap between PML- and p53-interacting human proteins. PML and p53 interactors were retrieved from the BioGRID database (June 2022). Of those interactors, 299 associate with both p53 and PML. A list of these proteins is given in the Table 1. (**b**) DAVID functional gene ontology (GO) enrichment analysis (June 2022) of the 299 proteins interacting with both p53 and PML. Different colors represent biological process, molecular function, and cellular component GO categories. The top 10 most significantly enriched GO terms for each category are plotted. DAVID, Database for Annotation, Visualization and Integrated Discovery; GO, gene ontology; neg., negative; pol, polymerase, pos., positive; TF, transcription factor.

**Figure 3 cancers-14-04549-f003:**
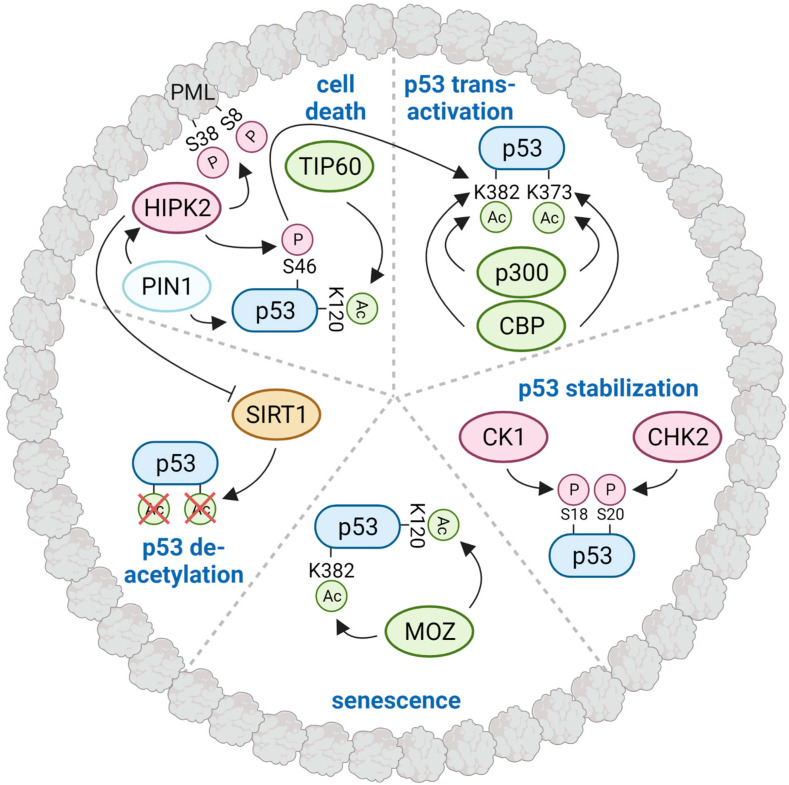
Control of p53 post-translational modifications and p53 activity at PML biocondensates. Schematic representation of a mature PML NB, in which PML resides at the peripheral shell. Phosphorylation of p53 at S18 and S20 by the kinases CK1δ/ε and CHK2, respectively, is involved in p53 stabilization. Acetylation of p53 at K373/K382 by the acetyltransferases p300 and CBP results in p53 transcriptional activation of target genes. p53 acetylation is counteracted by the deacetylase SIRT1, thereby inhibiting p53-dependent apoptosis, and facilitating cell survival after DNA damage. Interaction of p53 with the acetyltransferase MOZ at PML NBs leads to p53 acetylation at K120 and K382, and, subsequently, senescence induction. p53-dependent apoptosis is stimulated by TIP60-mediated acetylation of p53 at K120 and HIPK2-catalyzed phosphorylation of S46. The p53 S46 phosphorylation mark enables interaction of p53 with the prolyl-peptidyl cis/trans isomerase, PIN1, which catalyzes p53 isomerization. This conformational change stimulates p300 and CBP-mediated p53 acetylation, thereby promoting transactivation of cell death-stimulating p53 target genes. PIN1 is also important for DNA damage-induced HIPK2 activation. Upon lethal DNA damage, HIPK2 phosphorylates SIRT1 at S682 and inhibits SIRT1 activity, thereby increasing p53 acetylation and cell death. In addition, HIPK2 also regulates PML through phosphorylation at S8 and S38, which stimulates PML SUMOylation and stabilization upon DNA damage. To what extent p53 posttranslational modifications and function are regulated by LLPS remains currently unclear.

**Table 1 cancers-14-04549-t001:** List of 299 human proteins which interact with both p53 and PML. Proteins interacting with p53 or PML protein were retrieved from the BioGRID database (status: June 2022), and proteins associating with both p53 and PML were identified. Proteins are listed in alphabetical order.

ACACA	CUL1	HNRNPM	NCOR1	RFC4	TDP2
ACTG1	DAXX	HNRNPR	NEDD1	RNF125	TES
ADD3	DBN1	HNRNPU	NFATC1	RNF20	TET2
AHNAK	DCP1A	HOMER3	NFRKB	RPL11	TFCP2
ANKRD2	DCTN2	HSF1	NPM1	RPL5	TNRC6B
ANXA1	DDX3X	HSP90AB1	NR3C1	RRM2	TOP2B
ANXA2	DDX50	HSPA1A	NR4A1	RTN4	TOPBP1
APEX1	DGCR14	HSPA5	NUFIP2	RUNX2	TOPORS
ARID3A	DHX15	HSPA6	NUPR1	RUNX3	TP53
ARIH2	DIS3	HSPA8	PALLD	S100B	TP53BP1
ARNT	DNAJB1	HSPB1	PARK7	SAFB	TP63
ASF1A	DNM2	HTT	PARP1	SART1	TRIM24
ATRX	ECT2	IFI16	PC	SATB1	TRIM25
ATXN3	EEF1A1P5	ILF3	PCBP1	SBNO1	TRIM27
AURKA	EGLN3	JAK1	PCCA	SENP1	TRIM28
AXIN1	EHMT2	JUN	PER2	SEPT9	TRIM33
AZGP1	EIF3C	KAT5	PIAS1	SFPQ	TRIM66
BANP	EIF3F	KAT6A	PIAS2	SIN3A	TRIM69
BCL2	EIF3G	KIF20A	PIAS3	SIRT1	TRIML2
BCL6	EIF4B	KIF5B	PIAS4	SKI	TUBA1C
BCOR	EP400	KMT2A	PIN1	SKP1	TUBB
BHLHE40	EPB41L2	KPNA4	PIP	SLAIN2	UBA52
BLM	EPB41L3	LDHB	PKM	SLC1A5	UBC
BRCA1	ERCC3	LIG3	PLAGL1	SLC3A2	UBE2I
BRCC3	ERCC6	LIMA1	PLCG1	SMAD2	UBE3A
BRD1	EXOSC9	LMNA	PLEKHA4	SMAD3	UHRF1
BRD4	EZR	LMNB1	PLOD3	SMARCA4	UIMC1
BRD8	FAM50A	LYZ	PML	SMC5	UPF1
BUB3	FBXW7	MAGEA2	PMS1	SMTN	USP10
CALD1	FLNA	MAGED2	POLK	SNW1	USP11
CASP8	FOS	MAP1LC3B	PPARG	SP1	USP7
CCNT1	FOXK1	MAP4	PPARGC1A	SP100	VIM
CCT6A	FUBP1	MAPK1	PPP1R13L	SP3	WDR5
CCT8	FXR1	MAPK3	PPWD1	SPAG9	WRN
CDK1	GATAD2A	MAPK7	PRDX1	SPTA1	XAB2
CDK2	GATAD2B	MAVS	PRPF3	SQSTM1	XRCC1
CDK6	GTF2I	MDC1	PSMC3	STAT3	YAP1
CDK7	GTF3C4	MDM2	PSMD2	STX5	YEATS2
CDKN2A	H2AFX	MED1	PSME3	SUMO1	YTHDF2
CHD4	HADHB	MED23	PYHIN1	SUMO2	YTHDF3
CHD8	HBS1L	MIF	RAD51	SUMO3	YWHAZ
CHEK2	HCFC1	MTOR	RAD54L2	SUZ12	ZBTB16
CMTR1	HDAC1	MYC	RANBP2	SYNCRIP	ZBTB33
CORO7	HDAC2	MYH9	RANGAP1	SYNE2	ZBTB5
CREBBP	HDLBP	MYO6	RB1	TAB1	ZC3HAV1
CSDE1	HELLS	NAA40	RBCK1	TAF6	ZMYM2
CSNK1D	HIPK2	NAB1	RBX1	TAF9	ZNF148
CSNK2A1	HIST1H4A	NAP1L1	RDX	TARS	ZNF451
CSNK2B	HNF4A	NBN	RECQL	TCERG1	ZYX
CSTA	HNRNPK	NCOA2	RELA	TDG

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
