# Peer review of "Regulating the p53 Tumor Suppressor Network at PML Biomolecular Condensates"

_cancers, 2022, doi:10.3390/cancers14194549_

Round 1
Reviewer 1 Report
Regulating the p53 tumor suppressor network at PML biomolecular condensates
The review article aims to provide an overview on the molecular basis of the formation of PML
nuclear bodies (NBs), and the relationship of PML NBs with p53, including the mechanisms by
which p53 is recruited to and becomes compartmentalized with other recruited proteins at PML
NBs, and how these concentrated proteins affect p53-related pathways and cellular processes.
General comments
Biomolecular condensates is an important and fast-growing area of research, in particular the
discovery that dysregulation of biomolecular condensates is closely associated with cancer. Thus
the topic covered by this article is timely and of importance.
The problem with this article is that many aspects are not sufficiently discussed or covered, and in
some cases, recent findings are not included and hence the information provided in some areas is
outdated.
Most cited references are prior to 2021, with a rare few in 2021 & 2022. Given the rapid
advancement in this field, it is critical to provide the most updated discoveries and findings. Thus
authors should provide more recent studies either as examples within the relevant context or cited
literature to boost the knowledge contribution of this review article. At its current form, the
knowledge contribution of this article to the field is limited.
Specific comments:
1. In section 2 – authors should mention that PML NBs also act as storage compartments
regulating the availability of protein in the nucleus – in addition to their involvement in biological
processes.
2. In section 3:
a. It is recommended that authors include schematic diagrams of the structure of PML
protein scaffold and organization of PML NBs and images of PML NBs interspersed in
the chromatin to help readability.
b. For the formation of PML NBs, authors should include more recent findings, such as the
recent findings that zinc levels influences PML nuclear body formation through the
regulation of SUMO1-SIM interactions that are required for PML-NB and function.
c. There is no mention of the structural studies of PML, which should be included as its
oligomerization is critical for PML NBs biogenesis.
d. Authors only discuss proteins at PML NBs, what about RNAs which have been
discovered in the vicinity of PML NBs and reported to be involved in the PML NBs
biogenesis.
e. Studies showing that sumolyation of K616 on PML has also been reported (line 146)
3. In section 4:
a. It is more appropriate to say PML NBs have been shown to associate with the p53-gene
locus (line 250)
b. The interactions of PML and PML biocondensates with p53 (lines 271-283) should be
discussed within the context of malignant cells. Specifically, how these interactions
affects p53’s anti-cancer (apoptotic, anti-proliferative) activities should be clearly described. Also, what’s the relevance of these interactions with respect to p53
inactivation which occurs in majority of cancer cells?
Author Response
We thank the Reviewer for the feedback. In the following, we will address all the points raised by the reviewer.
General comments:
"Biomolecular condensates is an important and fast-growing area of research, in particular the discovery that dysregulation of biomolecular condensates is closely associated with cancer. Thus the topic covered by this article is timely and of importance.
The problem with this article is that many aspects are not sufficiently discussed or covered, and in some cases, recent findings are not included and hence the information provided in some areas is outdated.
Most cited references are prior to 2021, with a rare few in 2021 & 2022. Given the rapid advancement in this field, it is critical to provide the most updated discoveries and findings. Thus authors should provide more recent studies either as examples within the relevant context or cited literature to boost the knowledge contribution of this review article. At its current form, the knowledge contribution of this article to the field is limited."
Response:
In our review, we aim to provide an overview on the molecular basis of the formation of PML NBs and the relationship of PML NBs with p53. Because we intent to give an overview, we do not go into too many details for all points mentioned in our manuscript since this would unbalance the review into a more structural view, which, however, is not our scope. We think that providing an overview on the major principles and mechanisms underlying PML biocondensate formation provides balanced in-depth information for giving the reader insight into the covered topics. Along the suggestions of the reviewer, we added more references, which were published in the last five years. To the best of our knowledge, our review is the only one covering both PML NB biogenesis, especially the recently discovered contribution of liquid-liquid phase separation to PML NB formation, and the regulation of the tumor suppressor p53 at PML NB. Thus, we are convinced that our review is a relevant contribution to the p53 literature. Finally, we would like to point out that the initial description and numerous seminal findings on the function and structure of PML NBs are dated before 2021 and need to be adequately cited in our review. This does not mean that they are "outdated". With LLPS an interesting novel mechanistic principle entered the PML field. Its biological relevance as well as to what extend LLPS is involved in regulating biological processes through PML NBs (under physiological conditions) is still unclear and remains to be elucidated in the future.
According to the reviewers’ suggestions we have revised the review article and included additional more recent publications in several sections. In the following the detailed changes are given for all changes not outlined in our point-by-point responses to the specific comments:
Page 3:
“PML NBs have been shown to be involved in numerous cellular processes including protein modifications, especially protein SUMOylation [19], regulation of gene expression and epigenetic regulation [31,32], the DNA damage response [33-36], apoptosis [37], cellular senescence [38], and antiviral responses [39]. Additionally, PML NBs have been linked to the regulation of nuclear protein availability by serving as storage depots for proteins. Upon specific stimuli such as heat shock, cytokine signaling or genotoxic stress, proteins can be released from PML NBs making them available when required [23,40-43]. Finally, PML NBs also play a role in protein degradation and protein quality control via recruitment of SUMO-targeted E3 ubiquitin ligases [44,45] and by regulating the resolution of cytoplasmic stress granules [46,47], illustrating a molecular linker function between nuclear and cytoplasmic stress response pathways.” (page 3, lines 120-131)
Page 4:
„Using a proximity labeling mass spectrometry approach, 59 proteins were identified to interact with PML in a SUMO-dependent manner. Since this approach cannot discriminate between interactions mediated via covalent SUMOylation or non-covalent SUMO-SIM interactions [64], the detailed molecular mechanisms underlying individual interactions remains to be determined.” (page 5, lines 194-198)
Specific comments:
- In section 2 – authors should mention that PML NBs also act as storage compartments regulating the availability of protein in the nucleus – in addition to their involvement in biological processes.
Response:
We agree with the reviewer that we missed this important function. We now added the following sentences to the revised version:
“Additionally, PML NBs have been linked to the regulation of nuclear protein availability by serving as storage depots for proteins. Upon specific stimuli such as heat shock, cytokine signaling or genotoxic stress, proteins can be released from PML NBs making them available when required [23,40-43].” (page 3, lines 124-127)
2.a) It is recommended that authors include schematic diagrams of the structure of PML protein scaffold and organization of PML NBs and images of PML NBs interspersed in the chromatin to help readability
Response:
As suggested by the reviewer, we have added a new figure (new Figure 1) depicting the current knowledge about PML NB assembly. (page 5-6, line 213-224)
2.b) For the formation of PML NBs, authors should include more recent findings, such as the recent findings that zinc levels influences PML nuclear body formation through the regulation of SUMO1-SIM interactions that are required for PML-NB and function.
Response:
We think that the Reviewer refers to the following paper: Lussier-Price et al., “Zinc controls PML nuclear body formation through regulation of a paralog specific auto-inhibition in SUMO1”, Nucleic Acids Research, 2022.
The online version of this paper has been published just two days before we submitted our review and was therefore missing in our review. We have included the following phrase referring to this publication in the revised version of the manuscript:
“Recently, it has been shown that the interaction of a phosphomimetic PML SIM mutant with SUMO1 is blocked by the intrinsically disordered N-terminal region of SUMO1 (but not SUMO2). This inhibitory effect can be overcome by the addition of Zinc. This finding adds another layer of complexity on how cells can regulate PML NB biogenesis [71].” (page 5, lines 207-211)
Additionally, we have added to the revised version of the manuscript that the initial phase of PML NB biogenesis is driven by RING tetramerization and B1 box polymerization
“Specifically, it has been shown that initial formation of PML NBs depends on RBCC oligomerization mediated by tetramerization of the RING domain and oligomerization of the B1 box of the RBCC region [51,52].” (page 4, lines 147-149)
2.c) There is no mention of the structural studies of PML, which should be included as its oligomerization is critical for PML NBs biogenesis.
Response:
Although we do not elaborate specifically on structural studies of PML in our manuscript, we had already cited two publications [51, 70] which investigated the structure of PML. Following the suggestion of the reviewer, in the revised version of the manuscript, we have cited three more such studies [52, 69, and 71 (see above)].
2.d) Authors only discuss proteins at PML NBs, what about RNAs which have been discovered in the vicinity of PML NBs and reported to be involved in the PML NBs biogenesis.
Response:
We thank the reviewer for having brought up this point. We have included a small paragraph on this important point to our revised version.
“In contrast to other biomolecular condensates such as nucleoli and stress granules, which contain nucleic acids and in particular RNA, the data for PML NBs on this topic are conflicting. Whereas some studies show that PML NBs contain RNA [24-27], others show that PML NBs are devoid of chromatin and RNA, but that nascent RNAs accumulate in the vicinity of PML NBs [28,29]. While it is still under debate if PML NBs comprise nucleic acids, it is clear that they consist of multiple different proteins.” (page 3, lines 107-113)
To our knowledge, RNAs do not play a fundamental role in PML NB biogenesis. Thus, we have added the following sentences:
“In contrast to nucleoli, where RNAs have been shown to be involved in biocondensate formation [83,84], RNAs apparently do not play a fundamental role in PML NB biogenesis as transcription inhibition does not disrupt PML NBs [25,85]. Thus, PML NB formation seems to be driven by oligomerization of PML and SUMO-SIM-dependent recruitment of associated proteins with a likely contribution of LLPS to both processes.” (page 7, lines 289-293)
2.e) Studies showing that sumolyation of K616 on PML has also been reported (line 146).
Response:
We are grateful to the reviewer for this important comment: We have now added this information to our revised version:
“Additionally, PML has been found to be SUMOylated at K487 and K616 [59,60]. Employing mass spectrometry-based approaches further SUMOylation sites of PML, namely K380, K400, K460, and K497, have been discovered in recent years [61-63]. Although the detailed function of these modifications still remains elusive, they might regulate the recruitment of SIM-containing proteins to PML NBs.” (page 4, lines 164-168).
We have also corrected a typo in line 146 of the original manuscript. One of the SUMOylated lysine residues has been mislabeled as K56 instead of K65. Thus, we changed K56 to K65 in the revised manuscript (page 4, line 163).
3.a) It is more appropriate to say PML NBs have been shown to associate with the p53-gene locus (line 250).
Response:
Thank you. We changed this in line with the comment of the reviewer:
“PML NBs have been shown to associate with the p53 encoding gene locus” (page 7, line 298).
3.b) The interactions of PML and PML biocondensates with p53 (lines 271-283) should be discussed within the context of malignant cells. Specifically, how these interactions affects p53’s anti-cancer (apoptotic, anti-proliferative) activities should be clearly described. Also, what’s the relevance of these interactions with respect to p53 inactivation which occurs in majority of cancer cells?
Response:
We thank the reviewer for this interesting suggestion. We have added a paragraph describing the role of PML NBs for p53 signaling in the context of cancer with an emphasis on the little understood role of PML NBs in the context of p53 mutations. Thus, we changed the heading of section 4.2 to “PML biocondensates, p53 downstream responses and cancer” (page 12, line 485).
“PML and PML NBs are key mediators of p53 downstream tumor suppressive responses by regulating p53 stabilization, activation and p53 PTMs thus promoting p53-mediated apoptosis and cellular senescence upon DNA damage and aberrant onco-gene signaling. While the importance of PML NBs for wild-type p53 functions is undisputed, the role of PML NBs in regulating mutant p53 is currently unknown. p53 mutations, which are typically point mutations, are the most common mutations in human cancers with 35-42% of tumors harboring p53 mutations [133,134]. These mutations lead to a loss of p53 wild-type functions, but also often confer oncogenic gain-of-function (GOF) properties [100]. Although it has been shown that a p53 nonsense mutant can still localize to PML NBs [135] raising the possibility that other p53 mutants including the more common p53 missense mutations also localize to PML NBs, the functional consequences re-main unexplored. PML has been found to be important for cell proliferation, p53 transactivation [136] and STAT3 signaling in cancer cells harboring missense p53 mutations [137]. Interaction of mutant p53 with STAT3 has been shown to increase JAK2/STAT3 signaling promoting cancer cell proliferation and tumorigenesis [138]. Thus, it might be possible that association of mutant p53 with PML NBs mediates neomorphic GOF properties. Furthermore, it is also tempting to speculate that LLPS and/or deregulated LLPS events might contribute to the cellular phenotype and the carcinogenic function of cancer-derived p53 mutants.” (page 13, lines 498-516)
We thank the reviewer for providing constructive criticism that helped to improve the impact and quality of our review. article.

Reviewer 2 Report
This is a very comprehensive and timely review. Although it contains a lot of complexe information, it is well written and easily understandable. Overall, the authors have done a brilliant job.
The only thing I would criticize is the low number (only two) of figures. Eventually, the authors can further improve the review by adding two to three illustrating figures.
Minor:
Lane379: "PML NSs" should probably be "PML NBs"
Author Response
We are very grateful to the reviewer for the positive feedback and for finding our review “well written and easily understandable” although it contains “a lot of complex information”. In the following, we will address the two issues raised by the reviewer.
The only thing I would criticize is the low number (only two) of figures. Eventually, the authors can further improve the review by adding two to three illustrating figures.
Response:
As suggested by the reviewer, we have added a new figure (new Figure 1) depicting the current knowledge about PML NB assembly. (page 5-6, line 213-224)
Minor:
Lane379: "PML NSs" should probably be "PML NBs"
Response:
We thank the reviewer for detecting this typo. We have changed “PML NSs” to “PML NBs” (page 11, lane 430).
Round 2
Reviewer 1 Report
The authors have addressed most of the comments satisfactorily.